# Structural Characteristics and Phylogenetic Analysis of the Mitochondrial Genomes of Four *Krisna* Species (Hemiptera: Cicadellidae: Iassinae)

**DOI:** 10.3390/genes14061175

**Published:** 2023-05-28

**Authors:** Yanqiong Yang, Jiajia Wang, Renhuai Dai, Xianyi Wang

**Affiliations:** 1Provincial Key Laboratory for Agricultural Pest Management of Mountainous Region, Institute of Entomology, Guizhou University, Guiyang 550025, China; yangyanq12@126.com (Y.Y.); wjchuzhou@163.com (J.W.); 2Engineering Research Center of Medical Biotechnology, School of Biology and Engineering, Guizhou Medical University, Guiyang 550025, China; wangxianyi118@126.com

**Keywords:** Krisnini, mitogenome, codon usage bias, natural selection, phylogeny

## Abstract

*Krisna* species are insects that have piercing–sucking mouthparts and belong to the Krisnini tribe in the Iassinae subfamily of leafhoppers in the Cicadellidae family. In this study, we sequenced and compared the mitochondrial genomes (mitogenomes) of four *Krisna* species. The results showed that all four mitogenomes were composed of cyclic double-stranded molecules and contained 13 protein-coding genes (PCGs) and 22 and 2 genes coding for tRNAs and rRNAs, respectively. Those mitogenomes exhibited similar base composition, gene size, and codon usage patterns for the protein-coding genes. The analysis of the nonsynonymous substitution rate (Ka)/synonymous substitution rate (Ks) showed that evolution occurred the fastest in *ND4* and the slowest in *COI*. 13 PCGs that underwent purification selection were suitable for studying phylogenetic relationships within *Krisna*. *ND2*, *ND6*, and *ATP6* had highly variable nucleotide diversity, whereas *COI* and *ND1* exhibited the lowest diversity. Genes or gene regions with high nucleotide diversity can provide potential marker candidates for population genetics and species delimitation in *Krisna*. Analyses of parity and neutral plots showed that both natural selection and mutation pressure affected the codon usage bias. In the phylogenetic analysis, all subfamilies were restored to a monophyletic group; the Krisnini tribe is monophyletic, and the *Krisna* genus is paraphyletic. Our study provides novel insights into the significance of the background nucleotide composition and codon usage patterns in the CDSs of the 13 mitochondrial PCGs of the *Krisna* genome, which could enable the identification of a different gene organization and may be used for accurate phylogenetic analysis of *Krisna* species.

## 1. Introduction

Iassinae (Hemiptera: Cicadellidae) is one of the largest groups in the Cicadellidae family and has a global distribution that includes over 2000 known species that are currently assigned to 12 tribes and 155 genera [1,2,3,4]. In this family, many species are arboreal, while some live on shrubs. Krisnini, one of the twelve tribes of Iassinae, is known from the Oriental region of the Old World and the Caribbean region of the New World. *Krisna*, the genus of the tribe Krisnini, has only 39 species recorded in the world, of which 13 species are recorded from China [5]. The Cicadellidae family is phytophagous and is an important component in ecosystems. Because of their piercing–sucking mouthparts, some species damage agricultural and forestry commercial crops directly by sucking plant sap or indirectly by acting as vectors of phytopathogens, spreading viruses between plants, leading to crop stunting and even death. However, the biology and host plants of Krisnini are poorly known. Linnavuori and Quartau reported *Piper nigrum* L. as the host of *Krisna olivacea* [6], and *Krisna strigicollis* has been reported to harm *Camellia sinensis* L. Most previous studies on Iassinae relationships have focused on morphological characteristics and molecular fragments [2,7].

Mitochondrial genomes (mitogenomes) contain information that is crucial to molecular evolution, such as base compositional bias, codon usage, and substitution rates [8,9]. In addition, mitochondria are extensively used in the study of the origin of biological evolution and genetic diversity because of their rapid evolution, simple structure, low levels of recombination, and high genome copy numbers [10,11]. The research contents mainly include explaining the origin of species, exploring the phylogeny of insects, and revealing the geographical distribution of species polymorphism [12,13]. At present, mitogenomes have also been widely used in phylogenetic studies of the Cicadellidae family [14,15,16,17], and >200 complete or partial mitogenomes of Cicadellidae species have been stored in GenBank. Most of these species belong to the Deltocephalinae, Cicadellinae, and Typhlocybinae subfamilies; however, only three mitogenomes of Krisnini were retrieved from GenBank. Limited samples and molecular markers may hinder phylogenetic studies of Krisnini at various taxonomic levels. Consequently, sequencing the mitogenomes of *Krisna* can enrich the molecular database of Krisnini and may help enrich population genetics and phylogenetic studies regarding Cicadellidae. Moreover, characterizing the mitogenomes may provide a preferable comprehension of phylogenetic relationships among the Krisnini members.

The disparity in the frequency of synonymous codon usage when encoding DNA is known as codon usage bias (CUB). Codon bias is a common and complex natural phenomenon present in many organisms [18]. The analysis of different organisms showed that the CUB is related to genetic expression, and the rate of translation elongation and the overall functionality of any protein can reveal the genomic structure and evolution characteristics of organisms [19,20,21,22]. The selection process favors the specific codons that help in the accurate and efficient translation of highly expressed genes. CUB can be caused by factors such as gene function, restriction of gene composition, translation selection, protein secondary structure, natural selection, and mutation [23]. However, the main factors affecting CUB are natural selection and the mutation pressure of background nucleotide compositions [24]. Thus, because of CUB, codons are used with a higher frequency than other synonymous codons, thereby leading to adaptive evolution [25].

Our research was based on the complete mitogenomes of three *Krisna* species, *Krisna expansiva*, *K. furcata*, and *K. quadrimaculosus*, and a partial mitogenome of *K. nigromarginata.* The purpose was to increase the diversity of the mitogenomes of *Krisna*, strengthen the understanding of them, and provide data for phylogenetic studies of other subfamilies. We interpreted the mitochondrial structure of these four species, including the gene sequence, nucleotide composition, codon usage patterns, protein properties, and factors affecting CUB, and analyzed their molecular phylogenetic relationship. These results provide a new perspective for understanding the identification, phylogeny, and evolution of *Krisna* and its related species.

## 2. Materials and Methods

### 2.1. Sample Collection and DNA Extraction

The samples of adult insects were collected by netting or light trapping from Yingjiang County, Yunnan Province, China, 5 June 2019 (*K. expansiva* and *K. furcata*); Kuankuoshui National Nature Reserve in Guizhou Province, China, 29 June 2021 (*K. quadrimaculosus*); and Ying Ge Ridge National Nature Reserve, Hainan Province, China, 14 May 2021 (*K. nigromarginata*). The samples were stored in anhydrous ethanol and returned to the laboratory for anatomical identification. Total DNA was extracted from the insect abdomen using the DNeasy^®^ Blood & Tissue Kit (Qiagen, Germany). The purity and concentration of the extracted DNA were assessed using a Nanodrop 2000 spectrophotometer and 1% agarose gel electrophoresis. Extracted DNA was stored at −20 °C.

### 2.2. Sequence Processing and Analysis

Mitogenomes of each species were sequenced using the Illumina HiSeq 6000 s-generation sequencing platform at BerryGenomics (Beijing, China) with 150-bp paired-end reads. The average insert length was 350 bp, and 6 GB of clean data were obtained. Clean sequences of each mitogenome were assembled using Geneious Prime 2019.2.1 software [26] and were based on the mitochondrial reference sequence of *Idioscopus clypealis* (GenBank: NC_039642). The assembled sequences were annotated using the MITOS web server [27], with the genetic codon “invertebrate” selected. The 13 protein-coding genes (PCGs) were predicted by the ORF Finder in Geneious Prime using invertebrate genetic codes. The mitogenomic map and comparative analyses were performed using the CGView comparison tool (https://proksee.ca/projects/new) (accessed on 12 October 2022) [28]. Furthermore, the relative synonymous codon usage (RSCU) values and codon numbers were calculated by MEGA X [29]. Nucleotide diversity (Pi) values were determined using sliding window analysis (a sliding window of 200 bp and a step size of 20 bp) in DnaSP 5 [30]. In addition, the nonsynonymous mutation rates (Ka), synonymous mutation rates (Ks), and Ka/Ks ratio for the PCGs were calculated in DnaSP 5 [30]. Correlation analysis was performed using IBM SPSS version 21.0 software. Finally, the percentage of the overall nucleotide composition (A, T, G, and C) of each mitogenome was calculated. The values of GC skew and AT skew were calculated as follows [31]:AT skew=(A−T)/(A+T) and GC skew=(G−C)/(G+C)

### 2.3. Effective Number of Codons

The effective number of codons (ENC), which typically ranges from 20 to 61 [32], was used to identify the bias of gene-synonymous codon use. An ENC value of 20 represents the maximum bias, whereas an ENC value of 61 indicates no bias. Genes with ENC values <35 are suggested to have a substantial codon bias [33]. ENC values were obtained using the Condon W software.

### 2.4. Parity Rule 2 Bias Plot

The GC bias (G3/(G3 + C3)) and AT bias (A3/(A3 + T3)) values were used to create a graph of parity rule 2 (PR 2) to evaluate the influence of mutation and selection pressures [34]. In this plot, the coordinate of the center is 0.5, 0.5, which demonstrates an absence of bias between mutation and selection rates. However, the degree of deviation from the center, which is an unequal distribution of the nucleobase, may refer to the existence of biases for mutation and selection forces [35].

### 2.5. Neutrality Plot

A neutrality plot of GC12 (the average of GC1 and GC2) versus GC3 analysis was performed to determine the extent of mutation and selection forces in the CUB among the genes [36]. Selection forces do not contribute when a change occurs in the third codon position of a synonymous codon as the corresponding amino acid remains the same [37]. In the neutrality plot analysis, a regression coefficient of <0.5 shows that the influence of natural selection on codon preference is greater than the mutation pressure. Conversely, a regression coefficient close to 1 indicates that mutation pressure was the dominant effect [36].

### 2.6. Correspondence Analysis

Correspondence analysis (COA) is a multivariate statistical tool based on the RSCU values of the 13 PCGs. Past 4.09 software was used to determine the trend of codon usage distribution on axis 1 and axis 2 [38,39].

### 2.7. Grand Average of Hydropathy

The grand average of hydropathy (GRAVY) score is the sum of the products of the frequency of each amino acid and the corresponding hydropathy index of each amino acid, which determines the overall hydrophobic and hydrophilic nature of a protein. A positive value indicates that the protein is hydrophobic, whereas a negative value indicates that the protein is hydrophilic [40]. GRAVY was calculated using Galaxy (http://www.gravy-calculator.de/index.php, accessed on 8 March 2023) [41].

### 2.8. Phylogenetic Analysis

The phylogenetic tree was constructed based on the nucleotide sequence of mitogenomes of 90 leafhopper species belonging to 12 genera (Appendix A), with *Callitettix braconoides* (NC_025497), *Magicicada tredecim* (NC_041652), and *Tettigades auropilosa* (MG737767) as the outgroups. Data were downloaded from the National Center for Biotechnology Information website, and Geneious Prime 2019.2.1 software was used to extract 13 PCGs, *12s* rRNA, and *16s* rRNA from the datasets [26]. Then, the nucleotide sequences of the 93 species were aligned in batches using the MAFFT v7.313 algorithm [42] integrated into PhyloSuite v1.2.1 [43]; gaps and ambiguous sites in the alignments were then removed using Gblocks 0.91b [44] in PhyloSuite v1.2.1 [43]. MEGA X [29] was used to concatenate individual gene alignments. The following data were obtained: (1) PCG12, first and second codons of 13 PCGs with 7192 nucleotides; (2) PCG12RNA, including the PCG12 plus two rRNAs with 8654 nucleotides. The optimal partitioning scheme and nucleotide substitution model for Bayesian inference (BI) and maximum likelihood (ML) phylogenetic analyses based on the two different datasets were selected using PartitionFinder 2.1.1 [45], with the branch lengths linked, the Bayesian information criterion model, and the greedy search algorithm. The ML [46] and BI methods [47] were used to construct the ML and BI trees.

## 3. Results

### 3.1. Basic Features of the Mitochondria of the Krisna

In our study, all *Krisna* species contained a typical 37 genes (22 tRNA and 2 rRNA genes, and 13 PCGs) and a large noncoding region (control region), where 23 genes were located on the heavy strand (J-strand) and 14 on the light strand (N-strand) (Figure 1). The four mitogenome sequences of *K. expansiva* (OQ674152), *K. furcate* (OQ674153), *K. nigromarginata* (OQ674154), and *K. quadrimaculosus* (OQ674155) were closed-circular molecules, ranging from 14,442 bp (*K. quadrimaculosus*) to 15,334 bp (*K. expansiva*). The newly sequenced mitogenomes of the four species were consistent in length and gene order with those of the other previously sequenced Iassinae species [48]. The gene rearrangement phenomenon is not present in this genus. The nucleotide compositions of these mitogenomes are shown in Table 1. These *Krisna* species exhibited a heavy AT nucleotide bias (81.4%, 80%, 80.3%, and 80.2%), which is consistent with that of other leafhopper mitogenomes [49,50,51,52,53]. Moreover, these mitogenomes had a positive AT skew (0.13–0.22) and a negative GC skew (−0.15 to −0.21).

### 3.2. PCGs and Codon Usage

Among the 13 PCGs, the longest was *COI* and the shortest was *ATP8*. *ND4*, *ND4L*, *ND5*, and *ND1* were coded on the N-strand, and *COI*, *COII*, *COIII*, *Cytb*, *ATP6*, *ATP8*, *ND2*, *ND3*, and *ND6* were coded on the J-strand (Figure 1). All PCGs started with ATN (ATA, ATT, ATC, and ATG) and ended with TAA, TAG, or the incomplete codon T-, except for *ATP8*, which started with TTG. This atypical initial codon phenomenon is often observed in genes in other leafhopper mitogenomes, especially in *ATP8* [54,55,56,57,58].

The AT content of PCGs (78.6–80%) was slightly lower than that of the whole genome (80–81.4%), and the AT skew (−0.14 to −0.13) and GC skew (−0.03 to 0.02) of 13 PCGs were similar among the four *Krisna* species (Table 1). Figure 2 summarizes the RSCU in the PCGs. The result revealed that the codon usage of *Krisna* was remarkably alike among leafhopper mitogenomes [59,60]. The most frequently used codon was UUA (L), followed by UCA (S1), and the least frequently used codon was CUG (L) in the four *Krisna* species. The codon usage of PCGs can also reflect the preference of the mitogenomes for AT base usage. The RSCU values for *Krisna* indicated a pattern toward more A and T than G and C.

### 3.3. Control Region

The control region, also known as the A + T-rich region, is the longest non-coding region, and many genes are involved in mitogenic replication and transcription. In the leafhopper family, the biggest difference in the length and composition of the control region is the main reason for the differences in the mitogenomes of different species [52]. In the four *Krisna* species, the control region was located between *12S* rRNA and *trnaI.* Herein, the length of the control region of *Krisna* was 1108, 876, 470, and 224 bp for *K. expansiva*, *K. furcata*, *K. nigromarginata*, and *K. quadrimaculosus*, respectively. The AT-rich region had the highest AT content, with 91.3% in *K. expansiva*, 86.7% in *K. furcata*, 84.5% in *K. nigromarginata*, and 90.2% in *K. quadrimaculosus* (Table 1).

### 3.4. Mitochondrial Gene Variation in Krisna

Nucleotide diversity is usually used to identify regions with high nucleotide differences and can guide the selection of species- or group-specific markers for molecular evolution research, especially for taxonomic groups with high morphological similarities [61,62]. Thirteen PCGs in *Krisna* were examined for variation, and the nucleotide diversity of each PCG was determined using the sliding window approach. We found variable nucleotide diversity both within and among PCGs (Figure 3A). The average values of nucleotide diversity calculated for individual genes ranged from 0.1592 (*COI*) to 0.2523 (*ND2*); the most variable region (5815–6014) was in *ND2* (Pi = 0.34), whereas the most conserved fragment (681–880) was in *COI* (Pi = 0.10), as found in other leafhopper species [15].

We calculated Ka, Ks, and Ka/Ks values to compare the evolutionary patterns of the 13 PCGs (Figure 3B) and found that all PCGs in *Krisna* had Ka/Ks values < 1, indicating that they were under purifying selection. In the mitochondrial genome, the evolution rate of different PCGs differed, and *ND4* had the fastest evolution rate, whereas *COI* had the slowest rate and was relatively conserved, which is consistent with other findings [63,64,65]. The specific arrangement is: *ND4* > *ND4L* > *ND5* > *ATP8* > *ND2* > *ND1* > *ND6* > *ATP6* > *ND3* > *COIII* > *COII* > *CYTB* > *COI.*

### 3.5. Analysis of CUB

#### 3.5.1. Relationship between ENC and Compositional Attributes

The ENC values (Table 2) in the reported *Krisna* mitogenomes ranged from 37.79 to 38.35 and essentially showed no variation among the four species examined in this study, although they exhibited some bias in codon usage. In addition, we performed a correlation analysis between the ENC, the overall composition (A%, T%, G%, C%, and GC%), and the third position (A3%, T3%, G3%, C3%, and GC3%) of the codon (Appendix A). A significant correlation with a positive value among homogeneous nucleotides and a significant correlation with a negative value among most of the heterogeneous nucleotides were observed in several of the mitochondrial genes, suggesting that mutational pressure can affect the base composition bias of mitochondrial genes of the *Krisna* species [66].

We calculated the GCall and the CG content of the first, second, and third bases of the mitochondrial genes of the four *Krisna* species. We found that their contents were all <50% and that the CG contents in different positions were different. The content of the second base was the highest, and the content of the third was the lowest, showing: GC2 > GC1 > GC3. The low CG content indicated that codons ending in A/T bases were favored during the evolution of the *Krisna* mitogenomes, which was consistent with the overall trend of the complete mitogenomes.

#### 3.5.2. PR2 Plot Analysis

To investigate whether the codon bias was caused by mutation pressure or natural selection, the relationship between the G and C contents and between the A and T contents in the 13 PCGs was analyzed using the PR2 bias plot (Figure 4). The PCGs of the four *Krisna* species are irregularly distributed, and most of the genes are distributed in the upper left corner of Figure 4. This explained that the CUB of the gene codons that encode protein in the four *Krisna* species was influenced by natural selection and mutation pressure.

#### 3.5.3. Neutrality Plot

Previous studies showed that the presence of a significant positive correlation in mitochondrial genes indicates the role of mutation forces (GC mutation bias) throughout the codon positions [67,68] in these genes. However, we did not observe any significant positive correlation in the mitochondrial genes (Table 3), indicating a weak influence of mutation pressure in these genes. Furthermore, the regression coefficient was between −0.9996 and 0.7818; only the regression coefficient of *ND2* and *ND4* was >0.5, and the rest of the PCGs were <0.5 (Figure 5). Therefore, the use of these four *Krisna* species’ codons was more influenced by natural selection and less affected by mutations.

#### 3.5.4. Correspondence Analysis

Analysis of RSCU values in the mitogenomes is used to explore differences in codon usage between genes. In our analysis, the matrix containing all zero-row codons and the stop codon was removed, which we observed in the first major axis (F1) to account for 49.26% (*ATP6*), 50.11% (*ATP8*), 43.08% (*COI*), 50.40% (*COII*), 45.17% (*COIII*), 53.3% (*CYTB*), 50.52% (*ND1*), 60.98% (*ND2*), 57.39% (*ND3*), 56.27% (*ND4*), 42.25% (*ND4L*), 52.58% (*ND5*), and 49.28% (*ND6*), of all variants in the gene set; whereas, the second spindle (F2) accounted for only 30.12% (*ATP6*), 35.33% (*ATP8*), 33.83% (*COI*), 28.3% (*COII*), 30.44% (*COIII*), 32.78% (*CYTB*), 36.18% (*ND1*), 25.09% (*ND2*), 26.20% (*ND3*), 27.63% (*ND4*), 34.78% (*ND4L*), 26.69% (*ND5*), and 36.93% (*ND6*) (Figure 6). This leads to the first axis being the main contributor to codon bias.

We observed that most of the codons were plotted close to the axis and the distribution was concentrated, suggesting that the base composition of mutation bias is related to codon bias in these genes, supporting previous findings [69]. However, a discrete distribution of some gene codons was also observed, suggesting that other factors can influence the use of gene codons.

#### 3.5.5. Properties of Proteins

The amino acid compositions in the coding DNA strands (CDSs) of the mitochondrial 13 PCGs of the *Krisna* genome were calculated. The overall frequency of each amino acid used in the mitogenomes (Figure 7) revealed that leucine (Leu), phenylalanine (Phe), serine (Ser), isoleucine (Ile), and methionine (Met) were more abundant compared with other amino acids, with alanine (Ala), cysteine (Cys), aspartate (Asp), glutamate (Glu), and arginine (Arg) being the least abundant. Furthermore, Leu, Phe, Ile, and Met are all hydrophobic amino acids, while Cys, Asp, Glu, and Arg are hydrophilic amino acids. Hydrophobic amino acids are therefore preferred in the mitochondrial genome of the genus *Krisna*, and we concluded that the properties of amino acids influence the selection and thus the formation of codon bias. The GRAVY of all the PCGs was positive (Table 4), indicating that the mitochondrial proteins of the *Krisna* genome are hydrophobic while also indicating that more hydrophobic amino acids are present in the *Krisna* mitogenomes and that there is a preference for their use. This confirms the above judgment that protein properties will affect the formation of CUB.

### 3.6. Phylogenetic Analysis

Substitution saturation tests revealed that the two candidate nucleotide sequence datasets (PCG12RNG and PCG12) were not saturated, with the value of the substitution saturation index (Iss) obviously lower than the critical values (Iss.cSym or Iss.cAsym) (Table 5). This indicated that the concatenated data were suitable for further phylogenetic analysis (ML and BI). The phylogenetic tree topologies based on analyses of the two datasets showed that some branching relationships were recovered uniformly in the four trees, although the resulting topology was not exactly the same (Figure 8 and Appendix A).

In this study, each subfamily reverted to monophyly, which is consistent with several previous molecular phylogenetic studies [48,55,57,70,71,72]. Several subfamilies are very stable, such as Iassinae, which emerged as a sister group to Coelidiinae, and Ledrinae, forming sister groups with Evacanthinae; Membracoidae and Megophthalminae formed a sisterhood in one clade, supporting the proposition that Membracoidae is derived from the paraphyletic Cicadellidae, as has been shown in previous studies [73,74,75,76]. In our study, Deltocephalinae formed a single clade and tended to be placed at the bottom of the phylogenetic tree as a sister group to other leafhoppers. Our study also confirmed that the genus *Olidiana* is not monophyletic in the Coelidiinae and can be divided into three branches. The two species *Olidiana ritcheriina* and *O. ritcheri* were clustered closely with the genus *Taharana*. The remaining species were split into two clades: one included *O. longsticka* and *O. obliqua,* and the other included *O. tongmaiensis*, which supports Wang‘s conclusion [77]. Krisnini is the sister group of Batracomorphini and Hyalojassini. The genus *Krisna* is the sister of *Gessius* and was restored as a paraphyletic group. In phylogeny, the more samples that are used, the more reliable the results are, and the more reasonable the classification system that can be proposed. However, the sample size involved in this study is small, and the paraphyletic grouping of the *Krisna* species should be further verified in the future by increasing the sample size.

## 4. Discussion

Our study showed that the mitogenomes of the *Krisna* species are highly conserved in gene content, gene size, base composition, and codon usage. The *COI* gene is often used as a universal DNA barcode for species classification, species identification, and phylogenetic relationship assessments [78]. However, in this study, the *COI* gene was the least variable and slowest evolving gene among all the genes; in contrast, the *ND4* and *ND2* genes were the most variable and fastest evolving genes. Therefore, we believe that the *ND4* and *ND2* genes may be more suitable candidate markers for population genetic research and *Krisna* taxonomic identification. Previous studies have also found that *ATP8*, *ND6,* and *ND2* genes are more suitable as candidate markers for population genetic research and the classification and identification of Lamiinae taxa [79]. Ma et al. also suggested that *ND6* and *ND4* be used as potential DNA markers for species and population identification and proposed that the practicality of using low-variation *COI* genes as Lepidopteran barcodes needs to be carefully tested and revised [62]. Additionally, all PCGs’ Ka/Ks values were <1, illustrating that they are evolving under purifying selection and are suitable for investigating phylogenetic relationships within *Krisna*.

The codon usage analysis showed that *Krisna* species exhibited CUB to some extent. Furthermore, the PR2 plot analyzed the relationship between the G and C and A and T content in 13 PCGs and showed that the CUB of *Krisna* was affected by both natural selection and mutational pressure. The regression coefficient of the neutrality plot was <0.5, thereby indicating the critical role of natural selection over mutation pressure in the CUB of *Krisna*. This result is consistent with the regression coefficient value in silkworm mtDNA (r = 0.244, <0.5) [23] and further supports the main action of natural selection in CUB and the secondary action of mutation pressure in CUB. To determine the usage trend of PCGs in *Krisna*, we performed COA and showed that the codon usage of each gene was different. Previously, the COA of the silkworm mitogenomes showed that the contribution rate of the main axis (F1) to the total variation was 24.51%, and the contribution rate of the second axis (F2) to the total variation was 7.46% [80]. Another study reported that the codon position at the end of AT is closer to the main axis than that at the end of GC, indicating that the codon bias is relevant to the mutation bias of the composition constraint [23]. In addition, several genes are reported to be discretely distributed, indicating that natural selection may affect the CUB. Similarly, another report on PCGs in the silkworm mitogenome gives similar results to ours [35]. Protein composition and amino acid properties have little effect on the formation of codon preference in different groups of insects, and their hydrophilicity is positive, which may be to maintain their biological function [35]. In general, the influencing factors of CUB are mutation pressure and selection, of which selection is the dominant factor.

The phylogenetic tree shows that all the subfamilies are monophyletic. Xue et al. and Dietrich et al. classified Macropsini and Idiocerini into the Eurymelinae subfamily [75,81]. However, the phylogenetic relationships identified in our study do not support this inclusion. All datasets revealed the monophyly of the Iassinae and the sister groups with the Coelidiinae, and the relationships among the Iassinae tribes were similar to those reported in previous molecular phylogenetic analyses based on *28S*, *12S*, and *H3* gene regions [2]. The *Krisna* species reported in this article belong to the Krisnini tribe, the sister group of Batracomorphini and Hyalojassini. The *Krisna* genus has not been classified as monophyletic and can be divided into two branches. *K. nigromarginata* and *K. furcata* were clustered and closely grouped with the genus *Gessius*. The remaining species were split into one clade. Hence, we propose that *Krisna* is not monophyletic. In future studies, we hope to increase the mitogenomic data of the genus to confirm whether it is paraphyletic.

In summary, many factors can affect the synonymous CUB of organisms. For the *Krisna* mitogenome, natural selection was found to dominate the CUB, and we believe that mutation bias plays only a relatively minor role. Moreover, our study provides novel insights into the significance of the background nucleotide compositions and codon usage patterns in the CDSs of the 13 mitochondrial PCGs of the *Krisna* genome, which could enable the identification of a different gene organization and may be used for accurate phylogenetic analysis of the *Krisna*, although a study focused on this still needs to be conducted.

## Figures and Tables

**Figure 1 genes-14-01175-f001:**
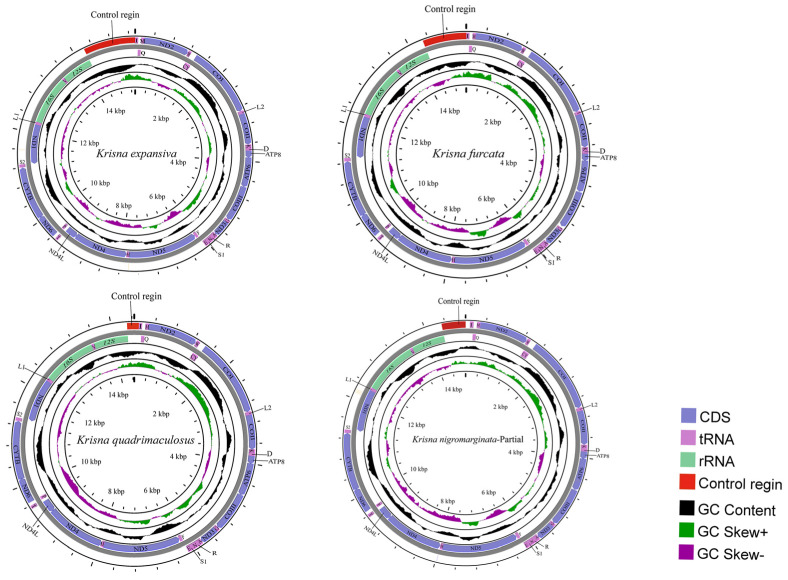
Circular map of the mitogenomes of four *Krisna* species.

**Figure 2 genes-14-01175-f002:**
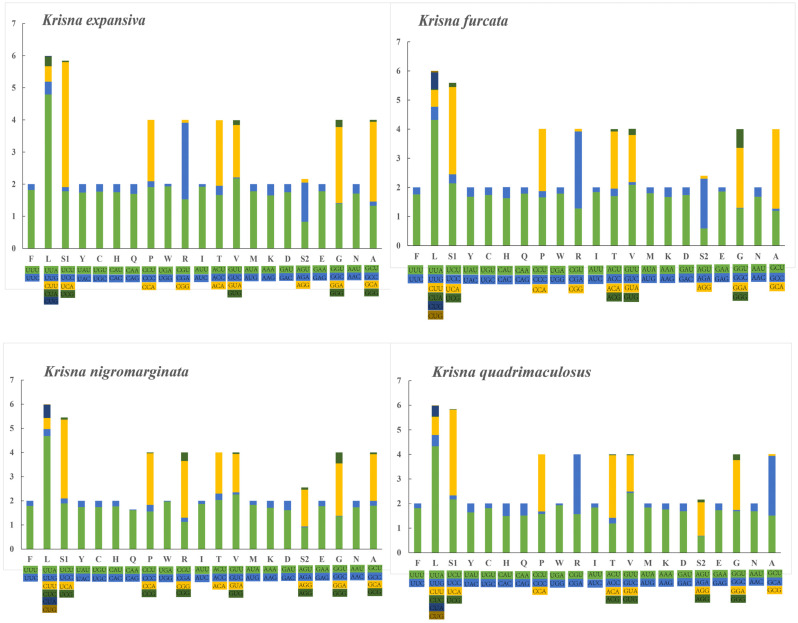
RSCU of the mitogenomes of the four *Krisna* species; the stop codon is not included.

**Figure 3 genes-14-01175-f003:**
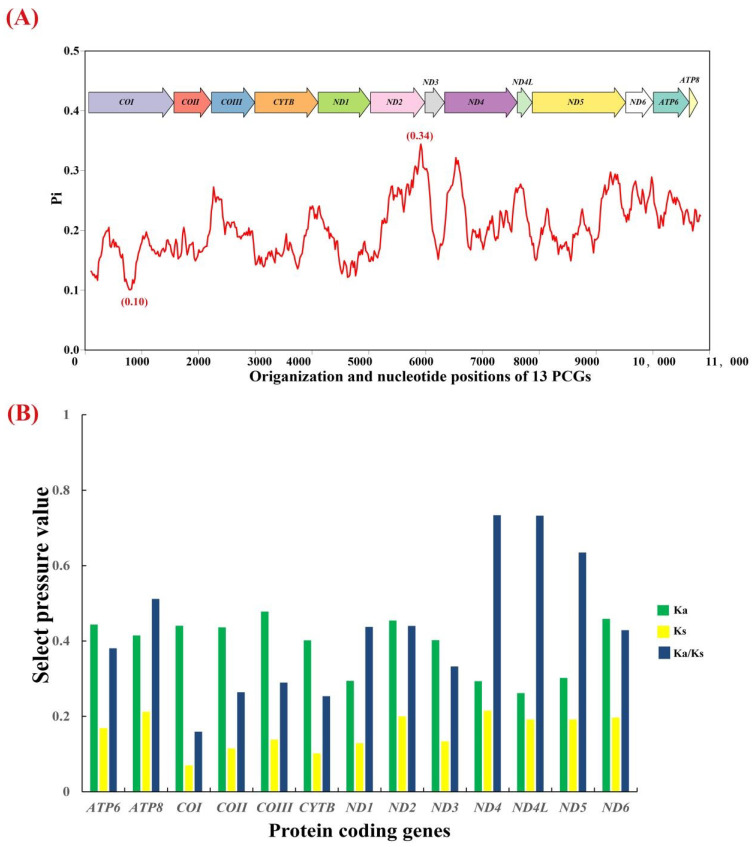
Gene variation in 13 PCGs in four *Krisna* species. (**A**) Sliding window analysis shows the value of nucleotide diversity. (**B**) Evolutionary rates of 13 PCGs.

**Figure 4 genes-14-01175-f004:**
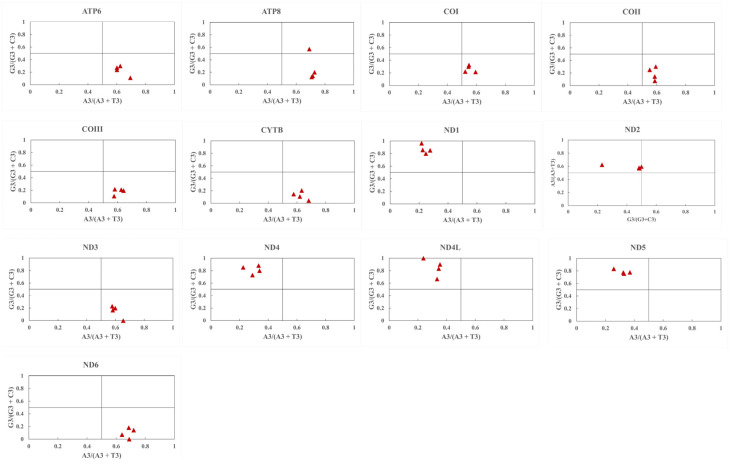
Parity plot analysis showing the proportionate distribution from the center that confirms the role of mutation pressure toward the codon usage. Values were taken for AT bias [A3/(A3 + T3)] along the X-axis and GC bias [G3/(G3 + C3)] bias along the Y-axis.

**Figure 5 genes-14-01175-f005:**
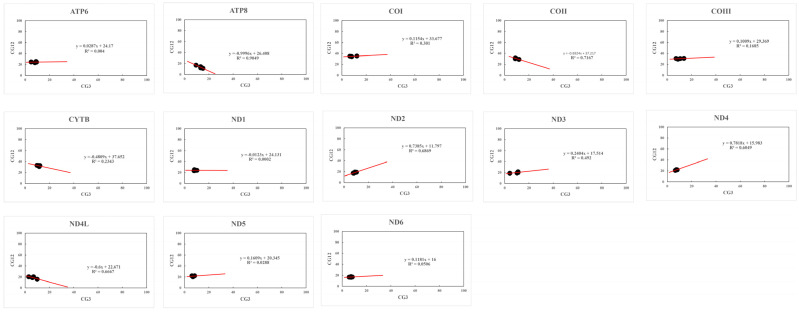
Neutrality plot of the PCGs of the mitogenomes of four *Krisna* species. GC3 and GC12 values are plotted on X and Y axes, respectively.

**Figure 6 genes-14-01175-f006:**
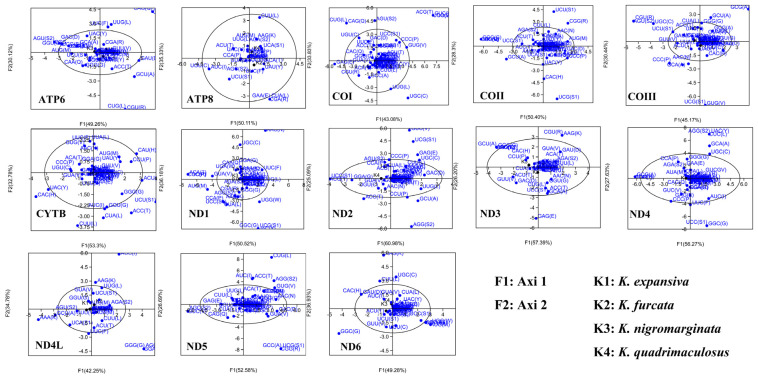
COA of the RSCU values for the mitochondrial genes of the four *Krisna* species. Blue dots indicate codons encoding amino acids in the genes.

**Figure 7 genes-14-01175-f007:**
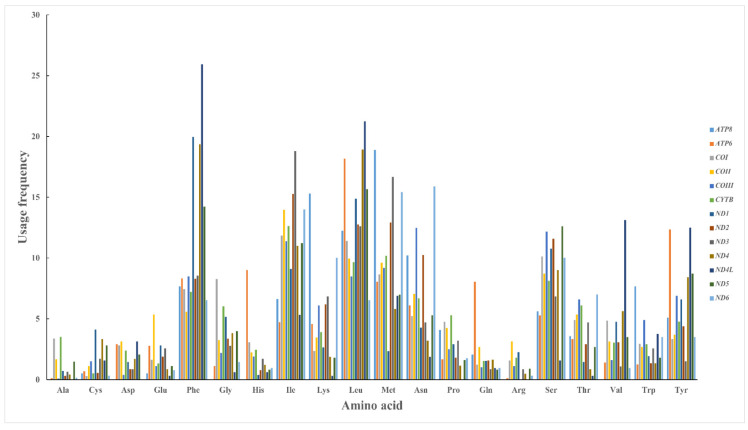
Overall amino acid usage frequency in the mitochondrial PCGs among the studied *Krisna* genomes.

**Figure 8 genes-14-01175-f008:**
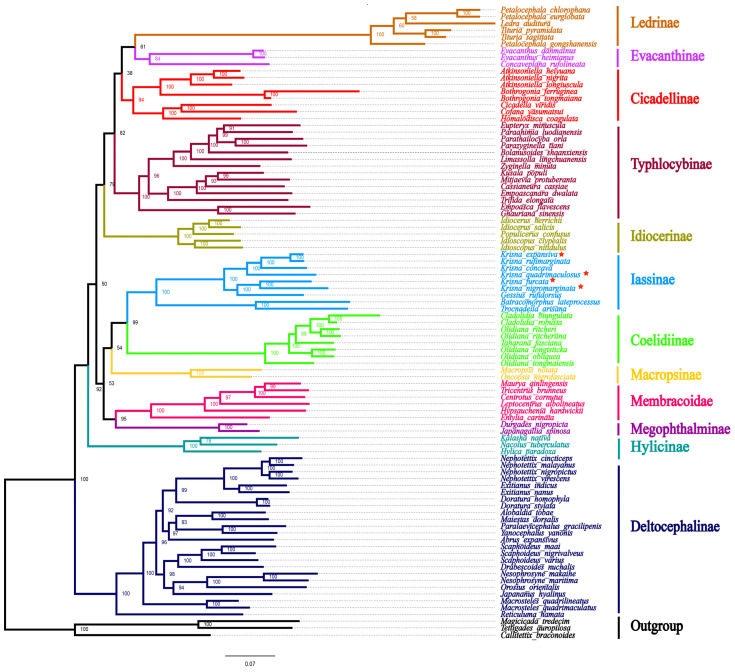
Phylogenetic tree of Cicadellidae species inferred via maximum likelihood analyses of the PCG12RNA; species with asterisks are the object of this study.

**Table 1 genes-14-01175-t001:** Nucleotide compositions of the complete mitogenomes of *Krisna* species.

Species	Feature	Length (bp)	A%	G%	C%	T%	AT Skew	CG Skew
*K. expansiva*	Whole mitogenome	15,334	46.9	7.9	10.6	34.5	0.15	−0.15
	Control region	1108	44.8	4.2	4.6	46.5	−0.02	−0.05
	13 PCGs	10,942	34.3	10.1	9.8	45.7	−0.14	0.02
	22 tRNAs	1423	40.6	11.0	7.0	41.3	−0.01	0.22
	2 rRNAs	1896	34.7	9.9	6.6	52.4	−0.26	0.20
*K. furcata*	Whole mitogenome	15,169	46.2	8.2	11.9	33.8	0.15	−0.18
	Control region	876	48.2	5.0	8.3	38.5	0.11	−0.25
	13 PCGs	10,943	34.3	10.6	10.8	44.3	−0.13	−0.01
	22 tRNAs	1430	41.3	10.8	7.6	40.3	0.01	0.22
	2 rRNAs	1918	30.9	10.7	6.5	51.9	−0.25	0.25
*K. nigromarginata*	Whole mitogenome	14,808	45.4	8.0	11.7	34.9	0.13	−0.19
	Control region	470	51.7	6.8	8.7	32.8	0.22	−0.12
	13 PCGs	10,952	34.4	10.2	10.2	45.1	−0.13	0
	22 tRNAs	1428	41.9	10.6	8.3	39.2	0.03	0.12
	2 rRNAs	1910	34.0	10.2	5.8	50.0	−0.19	0.27
*K. quadrimaculosus*	Whole mitogenome	14,442	49.1	7.8	12.0	31.1	0.22	−0.21
	Control region	224	55.4	4.9	4.9	34.8	0.23	0
	13 PCGs	10,936	34.7	10.2	10.6	44.8	−0.13	−0.03
	22 tRNAs	1424	41.6	11.0	7.4	40.0	0.02	0.20
	2 rRNAs	1903	28.3	10.6	6.8	54.3	−0.31	0.22

**Table 2 genes-14-01175-t002:** Codon features of PCGs of *Krisna*.

Species	ENC	GCall/%	CG1/%	CG2/%	CG3/%
*K. expansiva*	37.79	19.9	21.3	23.83	14.56
*K. furcata*	37.98	21.37	22.97	24.73	16.4
*K. nigromarginata*	37.94	20.45	22.45	24.27	14.64
*K. quadrimaculosus*	38.35	20.49	22.48	23.77	15.22

**Table 3 genes-14-01175-t003:** Correlation coefficient between CG12 and CG3.

ATP6:	0.348	CYTB:	−0.481	ND4L:	−0.351
ATP8:	−0.951 *	ND1:	0.067	ND5:	0.173
COI:	0.543	ND2:	0.445	ND6:	−0.328
COII:	−0.846	ND3:	0.701		
COIII:	0.438	ND4:	0.568		

* *p* < 0.05.

**Table 4 genes-14-01175-t004:** GRAVY of the 13 mitochondrial PCGs among four *Krisna* species’ genomes.

Species	*ATP6*	*ATP8*	*COI*	*COII*	*COIII*	*CYTB*	*ND1*	*ND2*	*ND3*	*ND4*	*ND4L*	*ND5*	*ND6*
*K. expansiva*	0.698	0.819	0.637	0.763	0.686	0.791	0.135	0.632	0.636	0.147	0.103	0.234	0.832
*K. furcata*	0.769	0.839	0.654	0.791	0.759	0.796	0.145	0.611	0.693	0.226	0.114	0.253	0.925
*K. nigromarginata*	0.706	0.894	0.617	0.763	0.714	0.784	0.160	0.584	0.681	0.218	0.073	0.305	0.821
*K. quadrimaculosus*	0.851	0.987	0.692	0.797	0.786	0.900	0.070	0.711	0.775	0.083	0.013	0.179	0.983

**Table 5 genes-14-01175-t005:** Substitution saturation tests for each dataset.

Dataset	NumOTU	Observed *Iss*	*Iss.cSym a*	*Psym b*	*Iss.cAsym c*	*Pasym d*
PCG12RNA	16	0.332	0.845	0.0000	0.679	0.0000
	32	0.344	0.816	0.0000	0.571	0.0000
PCG12	16	0.324	0.841	0.0000	0.681	0.0000
	32	0.330	0.814	0.0000	0.570	0.0000

a Critical values assuming a symmetrical tree. b Significant difference between Iss and Iss.cSym (two-tailed test). c Critical values assuming an extreme asymmetrical tree. d Significant difference between Iss and Iss.cAsym (two-tailed test).

## Data Availability

All mitogenome sequences generated in this study were deposited in GenBank under accession numbers OQ674152–OQ674155.

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
