# Peer review of "Structural Characteristics and Phylogenetic Analysis of the Mitochondrial Genomes of Four Krisna Species (Hemiptera: Cicadellidae: Iassinae)"

_genes, 2023, doi:10.3390/genes14061175_

Round 1
Reviewer 1 Report
1. Authors should revise the simple summary and Abstract make it in one paragraph with 200-250 words with the name of Abstract.
2. Add few recent article
3. In the Introduction section of the Ms is too short add 2 paragraph and references also.
4. In the material and methods section, provide the details information of sample collection (date, time etc.)
5. Figure 1 is not clear visible. Zoom the Image and high-resolution quality Image required.
6. The Discussions section need more references, integrated data and score subject wise.
7. The result is well written.
8. In Fig 2, kindly describe the legend.
9. In Fig 8, kindly describe the legend.
Conditionally recommended if authors performed all modifications clearly.
Minor editing of English language required
Author Response
Please see the attachment.
Thank you again for your advice!

Reviewer 2 Report
This paper reports the sequencing of complete mitochondrial genomes for 4 species of the cicadellid genus Krisna, adding to the two previously available mitogenomes for this genus. The authors summarize and illustrate overall characteristics of the sequenced genomes and conduct standard analyses of nucleotide composition, codon usage bias, synonymous codon usage, and hydropathy. They also conduct a phylogenetic analysis incorporating the new sequence data into a previous dataset comprising additional representatives of this particular tribe and subfamily as well as representatives of various other cicadellid subfamilies. The results are more or less consistent with previous results.
Overall the work is well done but it is not easy to determine how much these results add to our overall knowledge of leafhopper comparative genomics or phylogeny. Specifically, the authors compare some of the results of their analyses of mitogenome composition and evolution to that of the silkworm, Bombyx mori, but other measures (e.g., hydropathy) are not compared to other insects at all. In general, it would be much more meaningful to make such comparisons across the various major lineages of Cicadellidae. If there are systematic biases in any of the reported mitogenome composition measures, then this might help explain why the various previous phylogenetic analyses of this family based on mitogenomes (and other data) usually recover the same major lineages (subfamilies) as monophyletic, but disagree substantially in the relationships among these lineages.
The phylogenetic results are compared with those of several previous studies based on mitogenome data or data from partial genomes but it would also be useful to compare results for Iassinae to that of the most extensive previous molecular phylogenetic analysis of this group (Krishnankutty et al. 2016, Syst. Ent.)
The paper needs some additional editing to clarify the meaning of some statements. Examples:
Introduction:
“Krisnini, one of the 12 tribes of Iassinae, has only 39 species recorded globally[5].” Krisnini are not distributed globally; they occur only in the Old World tropics.
“Mitochondria are widely used in the study of the origin of biological evolution and genetic diversity because of their rapid evolution, simple structure, and effective genetic information[6-7].” The meaning of “effective genetic information” is unclear.
Discussion:
“Another study reported that the codon position at the end of AT is closer to the main axis than that at the end of GC, indicating that the codon bias is related to the mutation bias of the composition constraint[15].” The meaning of “codon position at the end of AT” and “…GC” is unclear.
“Ledrinae occupied the basal branch of leafhopper species in all phylogenetic analyses.” Figure 8 indicates that Ledrinae is sister to Evacanthinae on a relatively derived branch of the phylogeny subtended by most of the other included subfamilies.
Some careful additional editing of the English is needed to improve the diction and clarify some statements (see above).
Author Response
Please see the attachment. Thank you again for your advice. Best regards Sincerely

Reviewer 3 Report
Reviewers comments#
The manuscript addresses an interesting topic on “Structural Characteristics and Phylogenetic Analysis of the Mitochondrial Genomes of Four Krisna species (Hemiptera: Cicadellidae: Iassinae”which is quite interesting. The manuscript is well written. The work needs certain updates to improve the message and conclusion of the manuscript. The parameters chosen for strategy building are traditional and informative. The present form of the manuscript demands attention and clarifications in certain areas:
Abstract:
‘. our study provides novel ..” Please correct it to “Our study provides novel…”
“our study provides novel insights ……accurate phylogenetic analysis of the Krisna”.Too long sentence with less clarity and readability. Please split the sentence into two for a clear message.
Introduction: Why do you choose this insect? What type of damage does this insect portray for crops and what % of damage to economically important crops is reported (data)? Please write it in the introduction part to prove its importance for IPM.
What is the novelty and uniqueness that is special with your phylogenetic analysis please provide details. Please provide the latest references, which are at present missing.
Material and Methods: Section 2.1: Mention, how old are your collection and samples. Mention how insects were strored. What tissue was used for DNA extraction? Purification of the DNA might be a criteria prior to sequencing. Was it performed?
Section 2.2: Please provide extended details on hiseq which are missing in this section.
Section 3.3: Table legend it missing. It is fused with the text. The nucleotide compositions of the complete mitogenomes of Krisna species is not explaining important outcomes from this table. The text needs extension.
Fig 1 and Fig 4 are not clear/readable, please make it to publication quality.
Discussion: A few parts of the discussion are present in both the introduction and discussion sections. Please remove it from the discussion and provide an extended discussion with the latest updates compared with published references. Some key details related to mechanism and function are missing in the discussion. Please incorporate them for better understanding and elaboration?
References: Typo’s detected. “Dai: W.; Dietrich, C.H.; Zhang, Y.L. A review of the leafhopper “. Please check the references and correct them to the norms of the journal.
Conclusion:
The article provides interesting data but a few parts of the manuscripts are unclear. Since I found some degree of difficulty in reading and understanding certain parts of the manuscript, the article needs some corrections and clarifications as mentioned above. Importantly, some figures are not explained to its importance in the text and legends don’t explain the crucial information in the tables and figures. Altogether, I do think that the manuscript contains important issues, interesting approaches, and techniques, which can lead in understanding the accurate phylogenetic analysis of the Krisna with importance to the pest control strategies.
No comments
Author Response

(The authors gave the same response as above.)

Reviewer 4 Report
This manuscript aims to provide more molecular evidence to solve the phylogenetic problems in the tribe Krisnini. In the introduction, the Information on the essential phylogenetical background is very scant and lacks the study question, i.e. what is the hypothesis? The mere justification that the mitogenomes were not sequenced is not enough. Bioinformatically, the study was conducted correctly. The present article is well-prepared, but some inconsistencies in the discussion need explanation. Formally, the trees were generated correctly using the proper methods and an extensive sequence, which is correct from this point of view.
The right taxonomic Latin name should be used instead of treehopper.
Other comments are posted in PDF.

Author Response

(The authors gave the same response as above.)
